# Detection of Groundwater Levels Trends Using Innovative Trend Analysis Method in Temperate Climatic Conditions

**Ionuț Minea \*** [ID]**, Daniel Boicu and Oana-Elena Chelariu** [ID]

Faculty of Geography and Geology, Alexandru Ioan Cuza University of Iași, 700506 Iași, Romania; boicu.d.daniel@gmail.com (D.B.); oana.chelariu@uaic.ro (O.-E.C.)
\* Correspondence: ionutminea1979@yahoo.com; Tel.: +40-741-331-556

**Abstract:** The evolution of groundwater levels is difficult to predict over medium and long term in the context of global climate change. Innovative trend analysis method (ITA) was used to identify these trends, and ITA index was calculated to measure their magnitude. The data used are sourced from 71 hydrogeological wells that were dug between 1983 and 2018 and cover an area of over 8000 km$^2$ developed in the temperate continental climate in the north-eastern part of Romania. The results obtained by applying the ITA show a general positive trend for groundwater level over 50% of wells for winter and spring seasons and annual values. The negative trends were observed for more than 43% of wells for the autumn season followed by the summer season (less than 40%). The magnitude of trends across the region shows a significant increase for spring season (0.742) followed by winter season (0.353). Important changes in the trends slopes and magnitudes have been identified for groundwater level depth between 0 and 4 m (for winter and spring seasons) and between 4 and 6 m (for summer and autumn seasons). The results can be implemented in groundwater resources management projects at local and regional level.

**Keywords:** groundwater level; graphical method; innovative trend analysis (ITA); magnitude; north-eastern Romania

---

## 1. Introduction

Changes in hydro-climatic and hydrogeological conditions can cause significant disturbances in the natural elements that lead to difficult to predict trends. In this respect, it is a question of identifying methods that are relevant in the medium and long term, because of the lack of adequate monitoring of the hydro-climatic and hydrogeological parameters (air temperature, atmospheric precipitation, river flows, and groundwater level) in some parts of the world [1]. Moreover, the identification of trends in the variation regime of natural elements becomes relevant as the effect of climate change is felt increasingly globally [2]. In the past decades, the scientific research on the impact of climate change on the various hydro-climatic parameters has been multiplied, most of the analyses focusing on the identification of trends in the extreme seasonal and annual values [3,4]. The most commonly used statistical methods for identifying trends was the Mann Kendall test, Sen's slope, and linear regression [5–9]. These methods have also been applied to assess the impact of climate change on groundwater resources [10–12].

Recent scientific research proposes new statistical and graphical approaches to identify trends such as innovative trend analysis (ITA) [13]. This method was used in the analysis of the extreme values of rivers flow [14] or atmospheric precipitation [15,16]. The application of this statistical method is facilitated by the fact that comparative analysis of data series can be done without statistical assumptions such as common parametric and nonparametric trend tests [17–19]. For north-eastern

part of Romania previous researches based on Mann Kendall test and Sen's slope and linear regression have estimated significant changes in climatic [20,21], hydrological [22,23], and hydrogeological parameters [24] as a result of regional climate changes.

In this context, the main objective of this paper is the application of this novel method to identify the trends and magnitude of the trends of groundwater level under temperate continental climate conditions. This objective is important in the context of increasing impacts of climate change on water resources and can also be considered a starting point in the analysis of underground water resources evolution in close connection with the efficient management of these resources.

## 2. Data and Methods

### 2.1. Study Area

For the detection of groundwater level trends, data series from 71 hydrogeological wells included in 37 hydrogeological stations were used, covering an area of over 8000 km$^2$ in the north-eastern part of Romania (Figure 1). The drillings have been carried out since the 1980s, along Prut river hydrographic network and they are included in the Prut and its tributaries Groundwater body (GRPR02). This is an unconfined groundwater body developed on Sarmatian clay and sandy deposits [25].

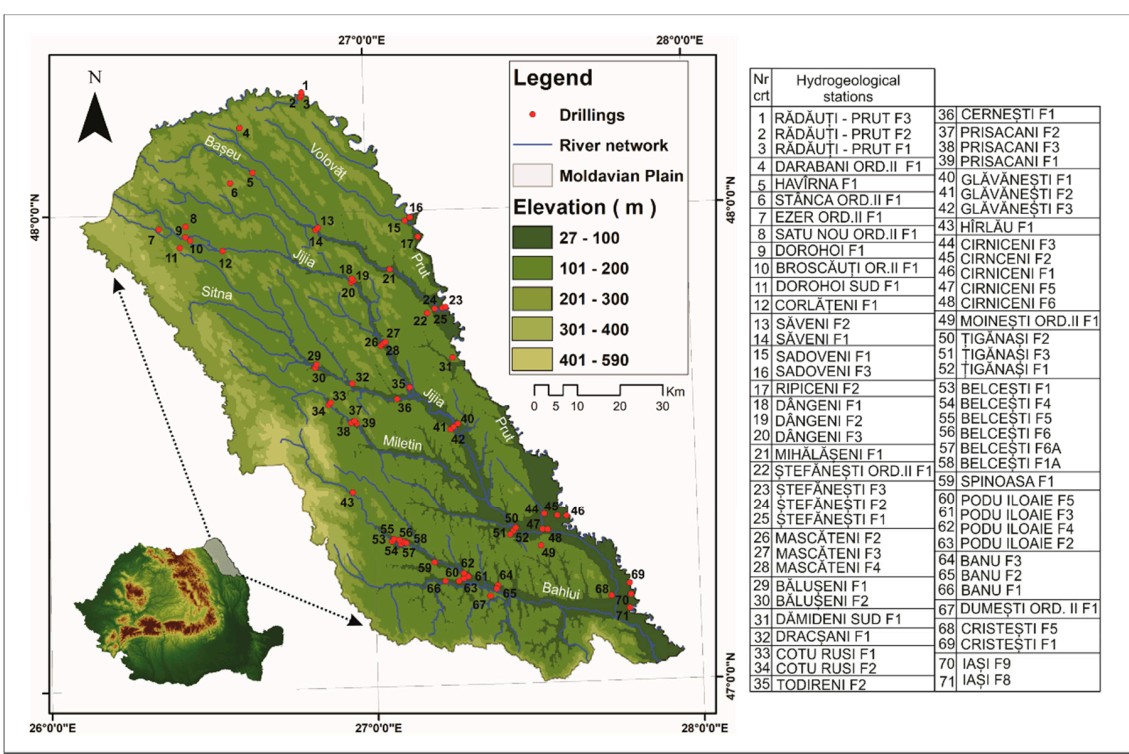

**Figure 1.** The distribution of wells in north-eastern part of Romania.

In terms of climatic conditions, this region is included in the temperate continental climate with maximum temperatures and rainfall in summer and minimum in winter. The whole region has a relatively homogeneous aspect in terms of natural conditions, with an average annual temperature varying between 8 °C (in the north part) and 9.5 °C (in the south part) and average annual rainfall between 550 mm (in the south) and 650 mm (in the north). In winter, because of negative average temperatures (from −6 °C to −4 °C) [26], precipitation is retained in the form of a snow layer. In this region, significant changes in climatic parameters [27] and hydrological [28] and hydrogeological ones [29] have been identified. Statistical analyses based on seasonal and annual average temperature

and rainfall data revealed that there is a positive trend for temperature and precipitation specific for spring and autumn, while decreasing precipitation characterizes the winter season [22].

## 2.2. Data

The statistical analysis was based on five time series data sets (four seasonal and one annual) for groundwater level from all 71 hydrogeological wells. The detection of trends was made only based on the series with data sets having no temporal interruptions over the period 1983–2018. The groundwater level data were provided by Prut-Siret Administration Branch of Romanian Waters Administration.

## 2.3. Innovative Trend Analysis Method (ITA)

This method has been applied for different extreme values of hydro-climatic parameters, considering that they have significant serial correlations at least on short memory basins [30]. In this paper, we apply this method to the data series regarding the groundwater level, considering that the measurements regarding the groundwater variation do not have the same accuracy and frequency as those related to the hydro-climatic parameters at the ground level. At the same time, it must be taken into account the fact that groundwater level variations are not as spectacular from day to day, sometimes even from one month to the next, so that the maximum and minimum values merge into average values measured at regular intervals of time. In the Romanian system, the measurements of the groundwater level are carried out every 3 days, and on average, in a month, ten measurements are made on a hydrogeological well. In particular situations, with constant climatic conditions (with alternating days with and without rainfall and without high temperature variations), the measured values do not differ greatly from one measurement to another. Therefore, the average monthly value fully reflects the natural variation of the groundwater level and may be considered as a basic element in the statistical analysis of trends using ITA.

In the first step, this method involves dividing the entire dataset into two equal and orderly series [31]. In the present study for the analysis of trends in the 1983–2018 time period, two sub-series of data, each of 18 years each, were extracted (1983–2000 and 2001–2018).

In stage two, the two sub-series are represented in a two-dimensional Cartesian coordinate system as scatter points and compared to the 1:1 (45°) line. After the graphical representation, if some of the points are above the median line, they show an increasing trend, while if they are below the median line, they represent a decreasing trend, and if they are concentrated along the median line they show no trend. Taking into account that in the case of groundwater level the actual measurements relate to the surface of the terrain, the interpretation of the trends is performed inversely to the interpretation of the trends in the hydro-climatic parameters from the surface (Figure 2).

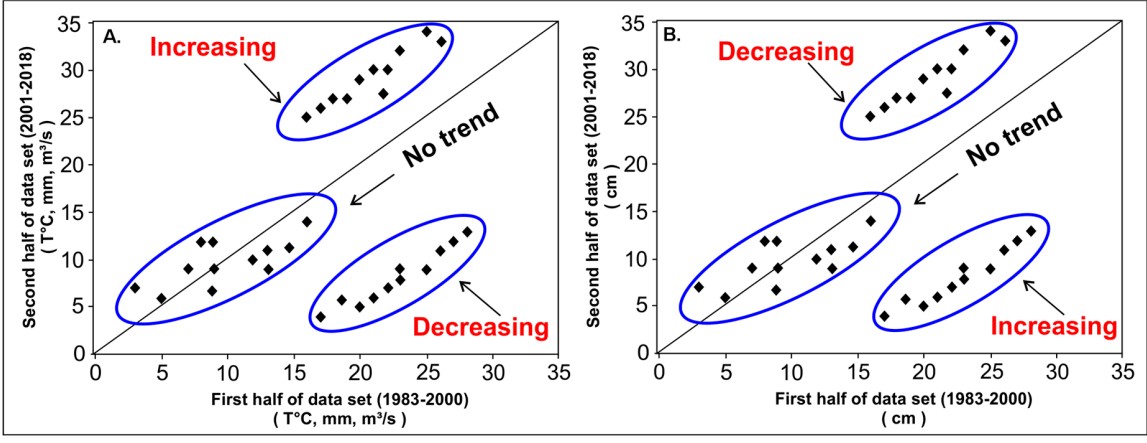

**Figure 2.** Graphical interpretation for climatic and hydrological parameters (**A**) and groundwater level (**B**) trends using innovative trend analysis (ITA).

Associated with this method, [13] also identifies a formula for estimating the slope:

$$S = \frac{2\,(\overline{y_2} - \overline{y_1})}{n} \tag{1}$$

where: S is the slope of trend, $\overline{y}_1$, $\overline{y}_2$ are the averages of the first and second series, and $n$ is the total number of the data. In the same time, the confidence limit (CL) of the trend slope must be taken into account. This can be expressed with the formula:

$$CL_{(1-\alpha)} = 0 \pm s_{crit}\sigma_s \tag{2}$$

where: $\alpha$ is the percent of significance level, $s_{crit}$ is standard deviation values [13] and $\sigma_s$ is the slope of standard deviation. For this study the significance for $\alpha$ is 5%. When the slope values are outside the confidence interval, the alternative hypotheses (Ha) is adopted. On this assumption, we can consider that there is a trend in the analyzed data. The type of trend is given by the slope sign, if it is negative the type of trend is increasing (for groundwater level case), if it is positive the type of trend is decreasing and if the slope is equal to 0 there is no trend.

At the same time, the magnitude of the trend can be calculated with ITA index [32] based on the formula:

$$ITA_i = \frac{1}{n}\sum_{i=1}^{n} \frac{10\,(y_i - x_i)}{\overline{X}} \tag{3}$$

where $x_i$ is the value of the first-ordered sub-series and $y_i$ is the value ordered of the second sub-series, $\overline{X}$ is the average of $x_i$.

In order to better highlight trends of variation in the level of groundwater for the entire region, the analysis was performed globally and on water depth classes. Five depth classes were achieved: the water depth class ranges between 0 and 2 m (including 12 wells), the water depth class ranges from 2 to 4 m (with 26 wells), the class with water depths of 4 to 6 m (with 14 wells), the water depth class range from 6 to 10 m (with 12 wells), and class with water depth in wells over 10 m (with 7 wells). This classification takes into account the mean depth of water and spatial distribution of the wells, most of the drillings being located along hydrographic network, the variation of the water level being influenced by a number of external factors such as the relationship between the groundwater level and the drainage of the rivers under drought conditions or high precipitations [33].

*2.4. Mapping*

For spatial interpolations for both slope and ITA index, the Ordinary Kriging (OK) method was chosen. In the analysis it was considered optimal to use this representation method because the geostatistical analysis revealed that the correlation coefficient between the measured values and the simulated values had the best results at the expense of other interpolation methods [34].

## 3. Results and Discussions

The ITA was applied to all wells and some of the results are shown in Figure 3. Because of the formatting constraints, we have represented in the present paper only 20 graphs resulting from applying this method (for Harlau F1, Banu, F2, Belceşti F1, and Podu Iloaiei F5) out of a total of 355 charts analyzed. The values of the statistical parameters associated with this method can be seen in Supplementary materials.

Taking into account the sign of slopes (positive or negative) across the region, the annual trends are positive for 54.9% of the wells and negative for 40.8% of the wells. No trend was recorded in 4.3% of drillings. Across seasons there are slight changes in the trend ratio. In winter, 52.2% of the wells registered a positive trend, 38.0% had a negative trend, and 9.8% had no statistically significant trend. The percentage of wells with positive trends has the highest value for the whole spring region (59.2%

of wells) because of increased surface water intake through the hydrographic network or infiltration of precipitation (Figure 4).

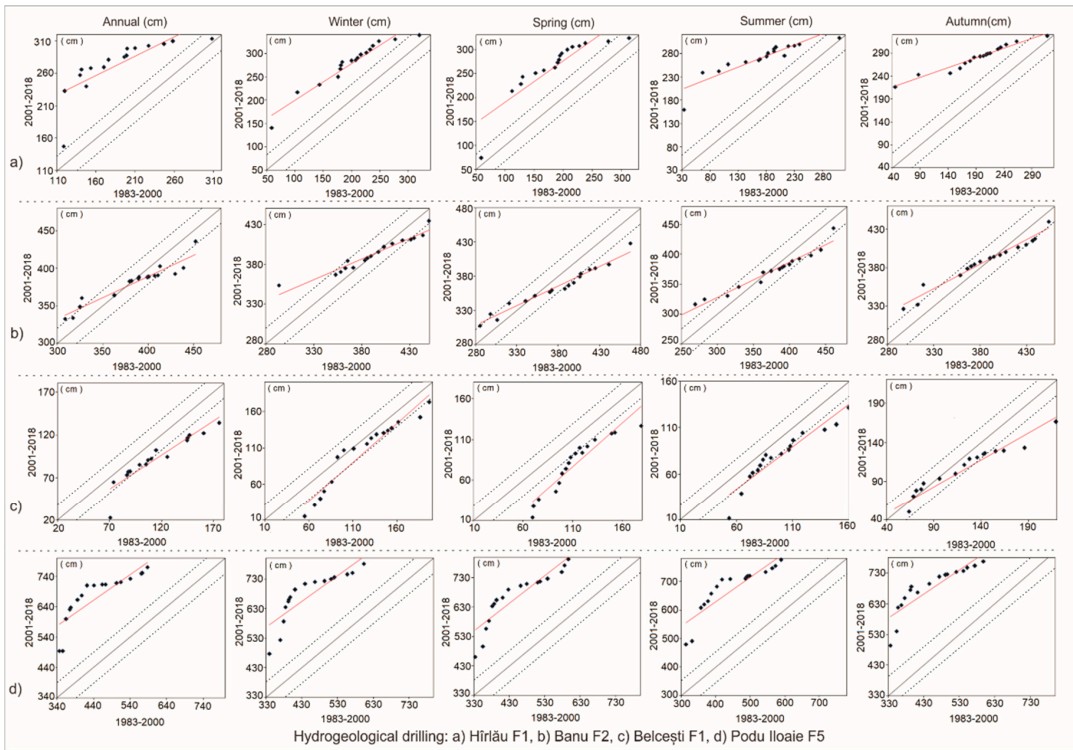

**Figure 3.** Trend analysis for season groundwater level based on ITA (**a**) Harlau F1, (**b**) Banu F2, (**c**) Belcești F1, (**d**) Podu Iloaiei F5.

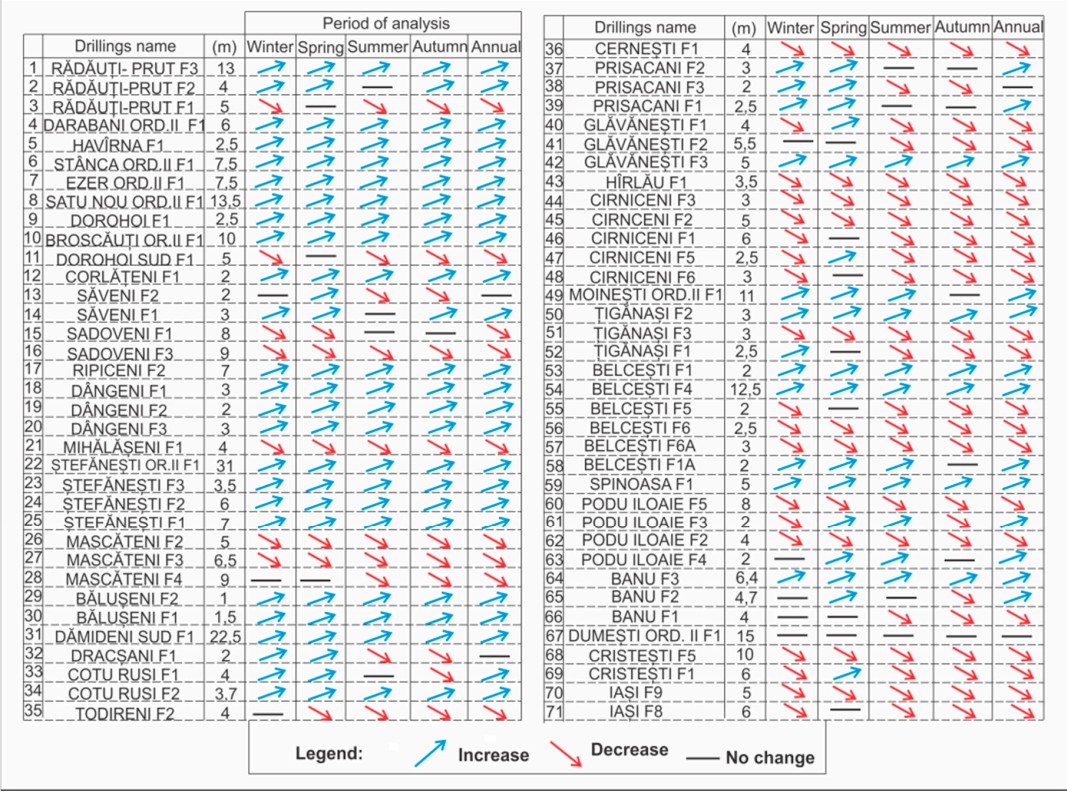

**Figure 4.** Seasonal and annual trends for groundwater level.

The lowest number of wells with negative trends (18 wells, representing 25.3% of the total) are registered in spring. A constant variation of the groundwater level is recorded during this season at 11 wells (15.5%). The lowest percentage of positive trend wells is registered in autumn (40.8%), and the highest percentage of wells with negative trends is recorded also in the autumn season (49.3%). The spatial distribution of slopes shows a general trend of growth in the north and north-west part and declines in the central and south-eastern parts (with small differentiations for the winter and spring seasons) (Figure 5).

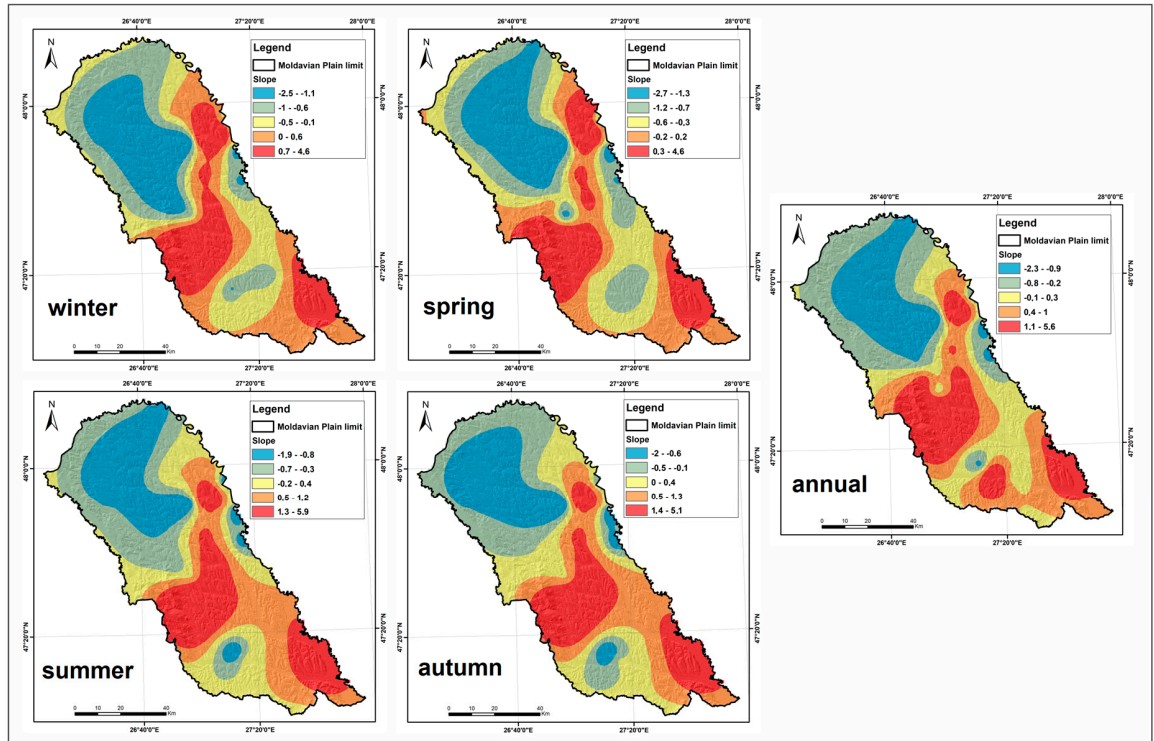

**Figure 5.** Slope distribution for annual and seasons periods.

On different depth of water levels, it is noted that for the depth class of 0 to 2 m, 91.7% of wells in summer season and 75% of wells for annual period record positive slopes. The highest percentage of wells with negative slopes is observed in the depth range of 4 to 6 m (with 71.4% of the wells) in the autumn season and in the winter and summer season (64.3% of the wells). Most wells where no trends have been found are in the depth class of 2 to 4 m (19.2% from wells in the summer season and 11.5% from wells in the spring season). Over 85% of the wells included in the depth class over 10 m have positive trends both for annual and all seasons (Figure 6). In fact, within this class there was no situation with negative trends of the slope, just for one well no trends were identified (Dumeşti F1) for all seasons.

The ITA index was applied to analyze the intensity of increase or decrease of groundwater level. For the entire region, the ITA index reveals a significant increase for the spring season (0.742) followed by the winter season (0.353). On an annual basis, the increase magnitude of the trends is slightly lower than that of the two season (0.291) because of the influence of the ITA index calculated for summer season (0.121) and autumn season (0.098) (Table 1). The values between +0.1 and −0.1 of the ITA index can be considered without significant statistical increase or decrease of groundwater level.

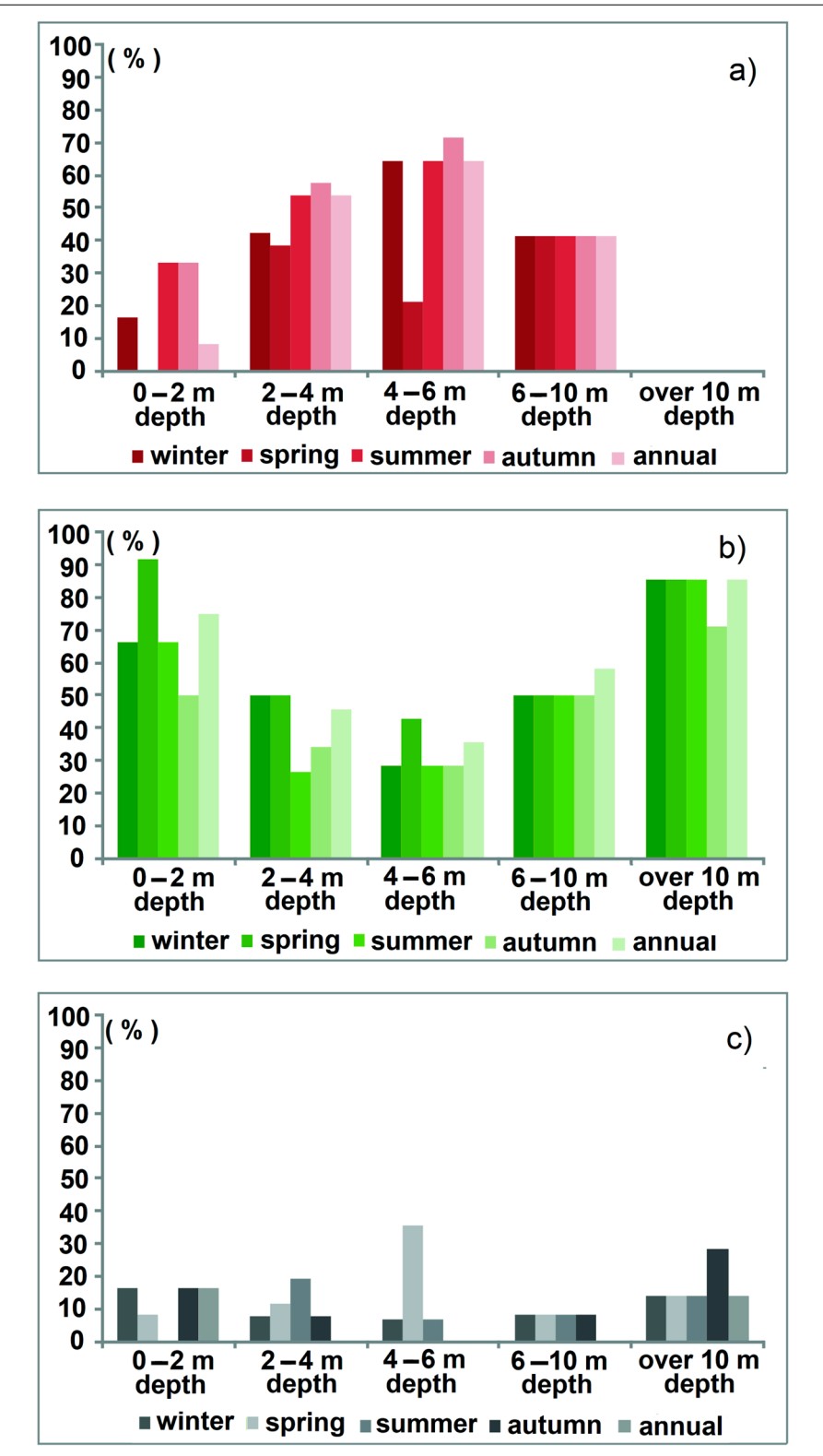

**Figure 6.** Positive (**a**), negative (**b**), and no trend (**c**) of slope frequency for different classes of groundwater level.

**Table 1.** Changes in seasons and annual groundwater level according to the ITA index.

| No. | Well Code | Average Depth (m) | Winter | Spring | Summer | Autumn | Annual |
|---|---|---|---|---|---|---|---|
| 1 | RĂDĂUȚI-PRUT F3 | 13 | 0.23 | 0.26 | 0.36 | 0.26 | 0.27 |
| 2 | RĂDĂUȚI-PRUT F2 | 4 | 0.12 | 0.48 | 0.07 | 0.12 | 0.19 |
| 3 | RĂDĂUȚI-PRUT F1 | 5 | −0.19 | 0.004 | −0.16 | −0.11 | −0.11 |
| 4 | DARABANI ORD.I F1 | 6 | 0.94 | 1.33 | 0.24 | 0.44 | 0.74 |
| 5 | HAVÎRNA F1 | 2.5 | 0.7 | 1.11 | 0.59 | 0.47 | 0.7 |
| 6 | STÂNCA ORD.II F1 | 7.5 | 0.61 | 0.79 | 0.7 | 0.5 | 0.65 |
| 7 | EZER ORD.II F1 | 7.5 | 0.34 | 0.46 | 0.14 | 0.24 | 0.29 |
| 8 | SATU NOU ORD.II F1 | 13.5 | 1.67 | 1.74 | 1.52 | 1.48 | 1.6 |
| 9 | DOROHOI F1 | 2.5 | 0.77 | 0.56 | 0.64 | 0.69 | 0.67 |
| 10 | BROSCĂUȚI ORD.II F1 | 10 | 1.53 | 1.64 | 1.82 | 1.63 | 1.65 |
| 11 | DOROHOI SUD ORD.II F1 | 5 | −0.068 | −0.087 | −0.4 | −0.2 | −0.18 |
| 12 | CORLĂTENI F1 | 2 | 1.4 | 1.75 | 0.82 | 0.66 | 1.12 |
| 13 | SĂVENI F2 | 2 | −0.018 | 0.33 | 0.42 | −0.26 | −0.12 |
| 14 | SAVENI F1 | 3 | 0.06 | 0.21 | 0.02 | 0.08 | 0.09 |
| 15 | SADOVENI F1 | 8 | −0.02 | −0.04 | −0.02 | −0.01 | −0.02 |
| 16 | SADOVENI F3 | 9 | −0.09 | −0.12 | −0.05 | −0.06 | −0.08 |
| 17 | RIPICENI F2 | 7 | 0.05 | 0.12 | 0.18 | 0.11 | 0.11 |
| 18 | DÂNGENI F1 | 3 | 0.59 | 1.06 | 0.39 | 0.32 | 0.57 |
| 19 | DÂNGENI F2 | 2 | 5.96 | 5.82 | 4.43 | 5 | 5.23 |
| 20 | DÂNGENI F3 | 3 | 1.28 | 1.79 | 1.1 | 0.94 | 1.25 |
| 21 | MIHĂLĂȘENI F1 | 4 | −1.06 | −1.01 | −1.15 | −1.03 | −1.06 |
| 22 | ȘTEFĂNEȘTI ORD.II F1 | 31 | 0.04 | 0.27 | 0.03 | 0.04 | 0.03 |
| 23 | ȘTEFĂNEȘTI F3 | 3.5 | 0.94 | 1.38 | 0.94 | 0.81 | 1.01 |
| 24 | ȘTEFĂNEȘTI F2 | 6 | 0.13 | 0.22 | 0.2 | 0.16 | 0.17 |
| 25 | ȘTEFĂNEȘTI F1 | 7 | 0.26 | 0.4 | 0.33 | 0.27 | 0.31 |
| 26 | MASCĂTENI F2 | 5 | −0.33 | −0.33 | −0.43 | −0.36 | −0.36 |
| 27 | MASCĂTENI F3 | 6.5 | −0.18 | −0.15 | −0.36 | −0.33 | −0.26 |
| 28 | MASCĂTENI F4 | 9 | −0.028 | 0.05 | −0.11 | −0.16 | −0.06 |
| 29 | BĂLUȘENI F2 | 1 | 4.25 | 12.1 | 1.77 | 1.13 | 2.93 |
| 30 | BĂLUȘENI F1 | 1.5 | 3.18 | 5.55 | 1.33 | 0.78 | 2.07 |
| 31 | DĂMIDENI SUD F1 | 22.5 | 0.05 | 0.05 | 0.05 | 0.05 | 0.05 |
| 32 | DRACȘANI F1 | 2 | 0.16 | 0.33 | −0.13 | −0.22 | 0.04 |
| 33 | COTU RUSI F1 | 4 | 0.31 | 1.7 | 0 | −0.47 | 0.24 |
| 34 | COTU RUSI F2 | 3.7 | 1.83 | 3.64 | 1.09 | 0.97 | 1.7 |
| 35 | TODIRENI F2 | 4 | 0 | −0.59 | −0.4 | −0.3 | −0.19 |
| 36 | CERNEȘTI F1 | 4 | −0.35 | −0.19 | −0.59 | −0.52 | −0.41 |
| 37 | PRISACANI F2 | 3 | 0.5 | 1.07 | 0 | 0.03 | 0.37 |
| 38 | PRISACANI F3 | 2 | 0.74 | 1.45 | −0.78 | −0.46 | 0.03 |
| 39 | PRISACANI F1 | 2.5 | 0.29 | 1.49 | 0.14 | −0.08 | 0.38 |
| 40 | GLĂVĂNEȘTI F1 | 4 | −0.14 | 0.28 | −0.28 | −0.53 | −0.19 |
| 41 | GLĂVĂNEȘTI F2 | 5.5 | −0.03 | 0 | −0.39 | −0.4 | −0.22 |
| 42 | GLĂVĂNEȘTI F3 | 5 | 0.3 | 0.5 | 0.27 | 0.17 | 0.31 |
| 43 | HÎRLĂU F1 | 3.5 | −1.3 | −1.4 | −1.8 | −0.5 | −1.5 |
| 44 | CIRNICENI F3 | 3 | −0.61 | −0.74 | −0.75 | −0.75 | −0.71 |
| 45 | CIRNICENI F2 | 5 | −0.45 | −0.51 | −0.49 | −0.56 | −0.5 |
| 46 | CIRNICENI F1 | 6 | −0.1 | −0.02 | −0.2 | −0.27 | −0.15 |
| 47 | CIRNICENI F5 | 2.5 | −0.35 | 0.18 | −0.53 | −0.88 | −0.41 |
| 48 | CIRNICENI F6 | 3 | −0.19 | 0.01 | −0.54 | −0.58 | −0.33 |
| 49 | MOINEȘTI ORD.II F1 | 11 | 0.04 | 0.05 | 0.05 | 0.01 | 0.04 |
| 50 | ȚIGĂNAȘI F2 | 3 | 4.86 | 5.18 | 2.63 | 2.7 | 3.57 |
| 51 | ȚIGĂNAȘI F3 | 3 | −0.96 | −0.9 | −1.34 | −1.25 | −1.12 |
| 52 | ȚIGĂNAȘI F1 | 2.5 | 0.21 | 0.19 | −0.85 | −0.67 | −0.34 |
| 53 | BELCEȘTI F1 | 2 | 0.9 | 2.1 | 1.2 | 0.2 | 1.2 |
| 54 | BELCEȘTI F4 | 12.5 | 0.07 | 0.1 | 0.1 | 0.02 | 0.08 |
| 55 | BELCEȘTI F5 | 2 | −0.18 | −0.02 | −0.51 | −0.44 | −0.29 |
| 56 | BELCEȘTI F6 | 2.5 | −0.9 | −0.7 | −1.1 | −0.2 | 0.9 |
| 57 | BELCEȘTI F6A | 3 | −0.6 | −0.4 | −0.8 | −0.3 | −0.7 |
| 58 | BELCEȘTI F1A | 2 | 0.38 | 1.5 | 0.57 | 0.05 | 0.6 |
| 59 | SPINOASA F1 | 5 | 0.21 | 0.13 | 0.3 | 0.12 | 0.23 |
| 60 | PODU ILOAIE F2 | 4 | −0.7 | 0.1 | 0.2 | 0.3 | 0.3 |
| 61 | PODU ILOAIE F3 | 2 | 0.16 | 0.13 | 0.12 | 0.1 | 0.08 |
| 62 | PODU ILOAIE F4 | 2 | 0.1 | 2.1 | 0.9 | 0.04 | 0.63 |
| 63 | PODU ILOAIE F5 | 8 | −1.63 | −1.57 | −1.64 | −0.5 | −1.62 |
| 64 | BANU F3 | 6.4 | 0.16 | 0.16 | 0.06 | 0.04 | 0.13 |
| 65 | BANU F2 | 4.7 | 0.02 | 0.2 | 0.03 | 0.013 | 0.04 |

<div align="center">**Table 1.** *Cont.*</div>

| No. | Well Code | Average Depth (m) | Winter | Spring | Summer | Autumn | Annual |
|---|---|---|---|---|---|---|---|
| 66 | BANU F1 | 4 | 0.03 | 0.06 | 0.27 | 0.05 | 0.1 |
| 67 | DUMEȘTI ORD.II F1 | 15 | 0.01 | 0 | 0 | 0 | 0 |
| 68 | CRISTEȘTI F5 | 10 | −0.41 | −0.39 | −0.43 | −0.45 | −0.42 |
| 69 | CRISTEȘTI F1 | 6 | −0.17 | −0.11 | −0.29 | −0.33 | −0.17 |
| 70 | IAȘI F9 | 5 | −0.17 | −0.18 | −0.51 | −0.46 | −0.33 |
| 71 | IAȘI F8 | 6 | −0.08 | 0.05 | −0.38 | −0.35 | −0.19 |
| | **Average** | | **0.35** | **0.74** | **0.12** | **0.10** | **0.29** |

For the depth class between 0 and 2 m of the groundwater level, 83% of the wells for winter season and 92% for spring season have positive values for ITA index (Figure 7). Most negative values of the ITA index were observed for the depth of groundwater between 4 and 6 m for summer and autumn seasons (64.3% for both). Important negative values for the ITA index were observed for the class between 2 and 4 m of the groundwater level with 42.35% of the wells in the spring season and yearly, and 46.2% of the wells in the summer and autumn season. For the wells with groundwater level over 10 m depths, the negative values for the ITA index were not calculated.

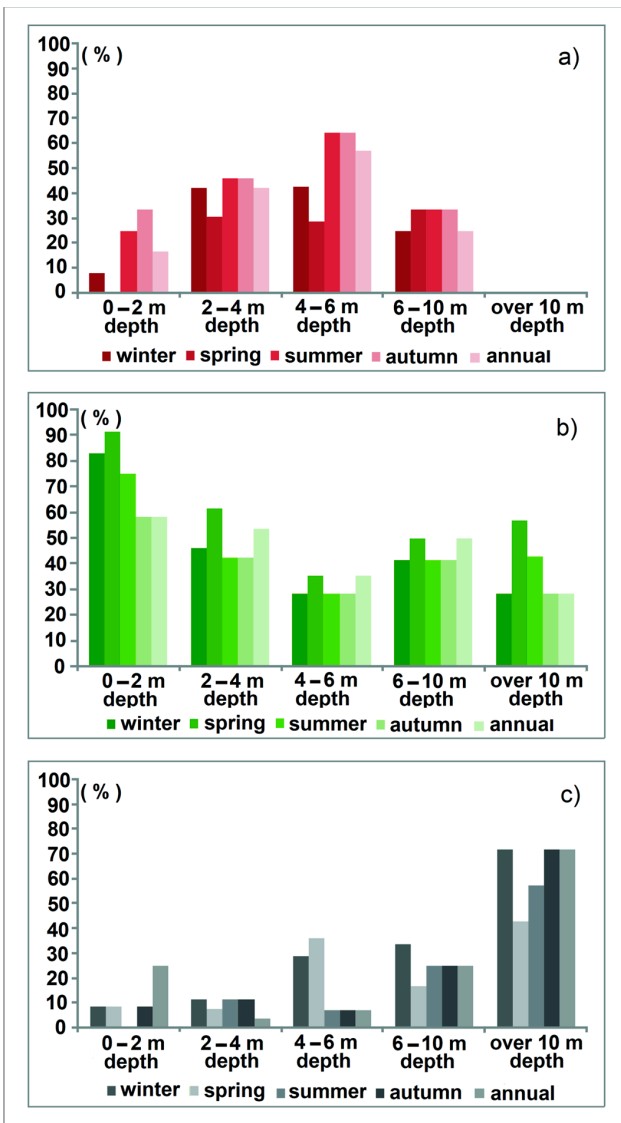

**Figure 7.** Frequency of positive (**a**), negative (**b**), and without statistical significance (**c**) of ITA index for different classes of groundwater level.

As we expect, most of the drilling where no significant values of ITA index have been observed, have the groundwater level at depths of over 10 m. At this level, 71.4% from seven wells included in this class have no significant values for ITA index in winter and autumn season and at the annual level, and 57.1 in summer season. This is determined by the high depth of the groundwater level and the reduction of the influence of surface water (from rivers, wetlands, and lakes).

For different classes of groundwater the values of ITA index records important variations. If for the winter and spring seasons the ITA index values remain positive for all depth classes, with a maximum for the 0 to 2 m class (1.42 and 2.76 respectively), for the summer and autumn there are negative values of the ITA index for the depth classes ranging from 2 to 4 m and 4 to 6 m, respectively (Figure 8). Statistical significant are those corresponding to the depth class of 4 to 6 m with ITA index for the summer season of −1.6. For annual values, the ITA index has significant positive values for depth classes ranging from 0 to 2 m (1.13) and over 10 m depth of groundwater level (0.48).

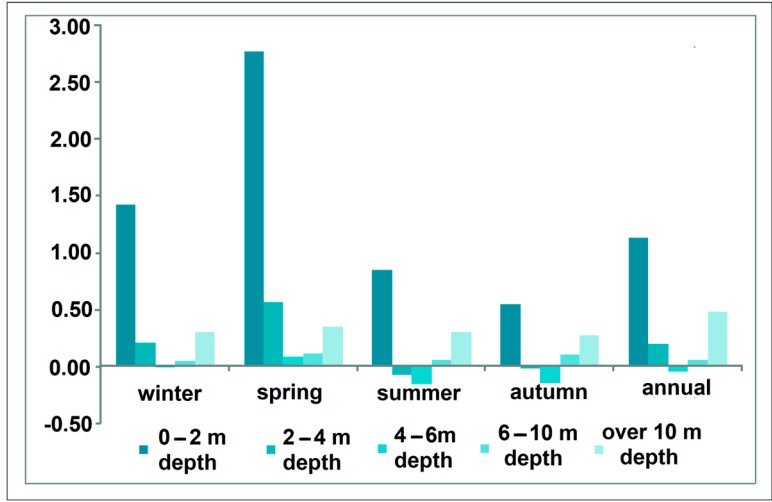

**Figure 8.** Changes in seasons and annual groundwater level according to the ITA index for different groundwater level depths classes.

These values of the ITA index are consistent with the slope trends identified for groundwater level for different classes. The same concordance is also observed in the spatial distribution of the ITA index where the lower values of the magnitude of the trend are recorded in the south-eastern, central, and south-western parts, and higher in the north and north-west (Figure 9).

These changes in the groundwater level can have a negative impact on the environment. In winter and spring seasons, the decrease in groundwater level, identified for depths of 0 and 2 m and depths of 2 and 4 m, can have a significant impact on surface water and underground water changes. Decreasing levels for groundwater may lead to lower groundwater quality through faster infiltration of surface water directly or through the hydrographic network [35,36].

Unlike classical methods that use the Mann-Kendall test, Sen's slope, and linear regression [37] to identify trends and their directions (increase, decrease, or constant), the ITA method can be applied to various hydroclimatic parameters. The comparative analysis of data series can be carried out without statistical assumptions [19] and the resulting graphs facilitate the statistical interpretation [38]. The method has been intensively used in the past decade to identify trends on extreme seasonal and annual value sets for climatic and hydrological parameters [39,40]. The ease of tracking trends by comparative analysis of data strings has led to the application of this method to identify solar radiation trends [41] and can be included in the analysis of trends of groundwater levels connected with different extreme hydroclimatic phenomena [42] or climatic changes [43].

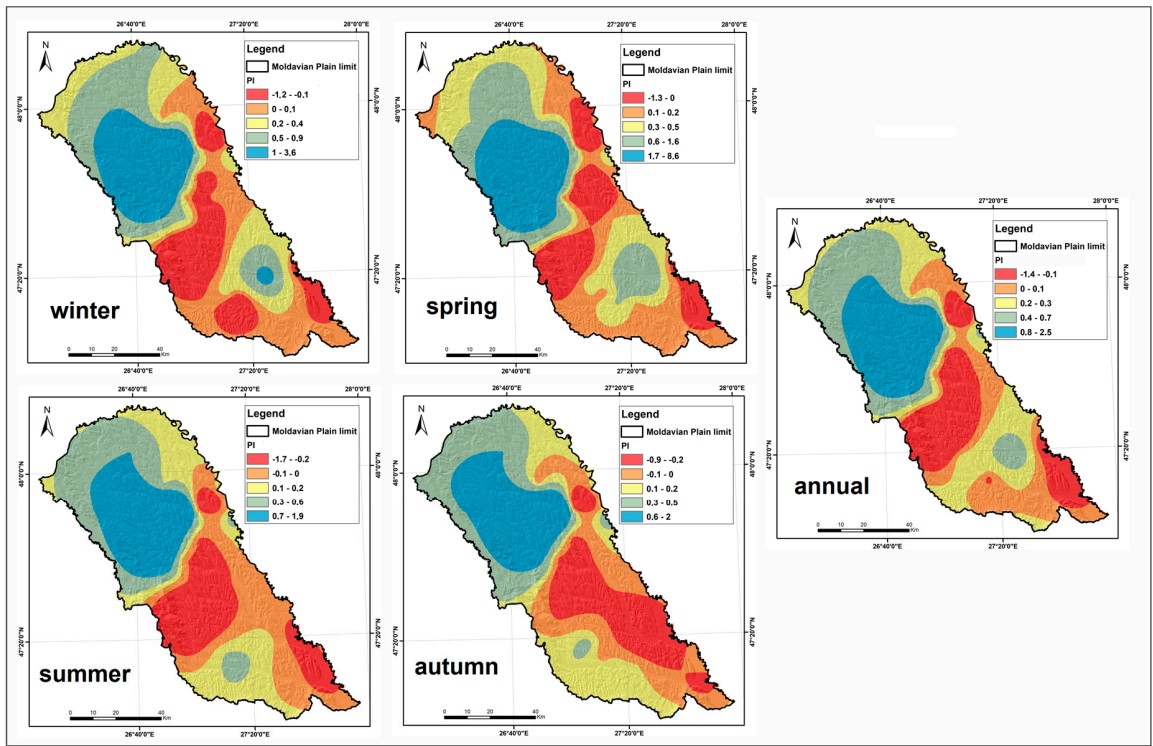

**Figure 9.** ITA index distribution for annual and seasons periods.

## 4. Conclusions

The main conclusions that can be drawn from the analysis performed for the assessment of groundwater level for wells trends in the north-eastern part of Romania can be summarized as follows:

1. The ITA reveals a general positive trend for groundwater levels over 50% of wells for annual, winter, and spring seasons. The negative trend for groundwater level was observed for more than 43% for the autumn season followed by summer season with less than 40%.
2. For different classes of groundwater level, in the winter and spring seasons, for the depth class of 0–2 m, positive trends were observed (with 83%, respectively 92% of the wells). For groundwater depths between 4 and 6 m, in the summer and autumn season, the most negative trends were observed (with over 64% of wells).
3. The ITA index indicates a significant positive increase of the groundwater level for the spring and winter season for the groundwater level depth between 0 and 2 m and 2 and 4 m. Significant negative values of the ITA index were observed for summer and autumn season for the groundwater class of 4–6 m.

Natural processes such as the earlier snowmelt, the shorter snow season, and the longer rainfall season are just a few elements that can substantially modify the hydro-climatic trends with effects on variation of the ground water levels. The results obtained by applying the ITA and associated index can be integrated into current and future local and regional management of underground water resources.

**Supplementary Materials:** The following are available online at http://www.mdpi.com/2073-4441/12/8/2129/s1.

**Author Contributions:** Conceptualization, I.M.; methodology, I.M. and D.B.; software, D.B. and O.-E.C.; validation, I.M.; formal analysis, O.-E.C.; investigation, D.B.; resources, I.M.; data curation, I.M.; writing—original draft preparation, D.B.; writing—review and editing, O.-E.C. and D.B.; visualization, I.M.; supervision, I.M.; project administration, I.M.; funding acquisition, I.M., O.-E.C. and D.B. All authors have read and agreed to the published version of the manuscript.

**Funding:** This research was funded by the grant of Minister of Research and Innovation CNCS—UEFISCDI, project number PN-III-P1-1.1-TE-2019-0286.

**Conflicts of Interest:** The authors declare no conflict of interest.

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
