# Peer review of "Detection of Groundwater Levels Trends Using Innovative Trend Analysis Method in Temperate Climatic Conditions"

_water, doi:10.3390/w12082129_

Round 1

Reviewer 1 Report

Interesting paper for the scientific community using innovative techniques. Recommended paper

Author Response

Thank you for your apreciation.

Reviewer 2 Report

The subject of the paper is interesting. Comments in attachment

Author Response

The subject of the paper is interesting. The authors have attempted to analyze an important problem of estimating trends in time series. ITA is one of many techniques proposed to detect deterministic trends in observed time series. This method attracted the attention of analysts as it was introduced with the appealing.

Having familiarize myself with the manuscript, I have a few comments

Thank you for appreciation. We tried to make all the corrections observed.

 Line 64 - should be (from -6oC to -4oC)

We made the modification required.

 Line 114-115 – How should be scrit and σs calculated? (provide a formula or references)

The most significant point in the application of this method is that the cross-correlation is between the

two sorted half time series. The statistical significance of the innovative trend slope test can be achieved

through a normal (Gaussian) PDF with zero mean and standard deviation equal to formula (Sen, 2017:

where n – number of data,

σ – standard deviation

ρ - correlation coefficient between the two mean values

   and are the arithmetic averages of the first and the second halves of the dependent variable.

In the text we chose to make the reference to Sen (2017) article.

 Line 137 – no Appendix I in text

Because the results obtained have more than 17 pages for 71 wells and 355 graphs, we chose to present in the paper just only 20 graphs resulting from applying of this method.

The appendix I are attached to this file and was attached to this paper as additional material at the time of initial submission.

 Line 140-148 - In my opinion this part of study should be placed in section: Material and methods

We moved this part at the end of data and methods chapter

 The papers results are quite interesting. However, the discussion section is poor and should be improved.

Thank for your observations. We tried to improve the discussion section and to connect our results to other researches base on this method. We added the text below (with references) to improve the discussion section.

Unlike classical methods that use the Mann-Kendall test, Sen’s slope and linear regression [37] to identify trends and their direction (increase, decrease or constant), the ITA method can be applied to various hydroclimatic parameters. The comparative analysis of data series can be carried out without statistical assumptions [19] and the resulting graphs facilitate the statistical interpretation [38]. The method has been intensively used in the last decade to identify trends on extreme seasonal and annual value sets for climatic and hydrological parameters [39, 40]. The ease of tracking trends by comparative analysis of data strings has led to the application of this method to identify solar radiation trends [41] and can be included in the analysis of trends of groundwater levels connected with different extreme phenomenon [42] or climatic changes [43].

[37] Bürger, G. On trend detection. Hydro Proc, 2017, 31, 4039–4042. doi.org/10.1002/hyp.11280

[38] Guslu, Y.S. Improved visualization for trend analysis by comparing with classical Mann-Kendall test and ITA, J Hydroloy, 2020, 584;  doi.org/10.1016/j.jhydrol.2020.124674

[39] Alifujiang, Y.;  Abuduwaili , J.; Maihemuti , B.;, Emin, B.; Groll, M. Innovative Trend Analysis of Precipitation in the Lake Issyk-Kul Basin, Kyrgyzstan, Atmosphere 2020, 11(4), 332; doi.org/10.3390/atmos11040332.

[40] Caloiero, T. Evaluation of rainfall trends in the South Island of New Zealand through the innovative trend analysis (ITA). Theor. Appl. Climatol. 2019, 139, 493–504. doi:10.1007/s00704-019-02988-5.

[41] Zhou, Z.; Wang, L.; Lin, A.; Zhang, M.; Niu, Z. Innovative trend analysis of solar radiation in China during 1962–2015. Renew. Energy 2018, 119, 675–689

[42] Halder, S.; Roy, M.B.; Roy, P.K. Analysis of groundwater level trend and groundwater drought using Standard Groundwater Level Index: a case study of an eastern river basin of West Bengal, India. SN Appl. Sci., 2020, 2, 507doi.org/10.1007/s42452-020-2302-6.

[43] Li, H.; Lu, Y.; Zheng, C.; Zhang, X.; Zhou, B.; Wu, J. Seasonal and Inter-Annual Variability ofGroundwater and Their Responses to Climate Change and Human Activities in Arid and Desert Areas: A Case Study in Yaoba Oasis, Northwest China, water, 2020, 12, 303; doi:10.3390/w12010303.

Reviewer 3 Report

I found the manuscript interesting and well prepared. The study is focused on assessment of groundwater level trends in north-eastern Romania. The comprehensive interpretation can be interesting to the readers from different countries. I just suggest modifying the beginning of the abstract because you are not analyzing hydrogeological parameters but groundwater levels. Hydrogeological parameters are usually considered to be hydraulic conductivity, transmissivity, storage ....

Author Response

Thank you for your appreciation. We made the correction in the abstract.